# Learning Pareto-Optimal Pandemic Intervention Policies with MORL

## Abstract

The COVID-19 pandemic underscored a critical need for intervention strategies that balance disease containment with socioeconomic stability. We approach this challenge by designing a framework for modeling and evaluating disease-spread prevention strategies. Our framework leverages multi-objective reinforcement learning (MORL)—a formulation necessitated by competing objectives—combined with a new stochastic differential equation (SDE) pandemic simulator, calibrated and validated against global COVID-19 data. Our simulator reproduces national-scale pandemic dynamics with orders of magnitude higher fidelity than other models commonly used in reinforcement learning (RL) approaches to pandemic intervention. Training a Pareto-Conditioned Network (PCN) agent on this simulator, we illustrate the direct policy trade-offs between epidemiological control and economic stability for COVID-19. Furthermore, we demonstrate the framework's generality by extending it to pathogens with different epidemiological profiles, such as polio and influenza, and show how these profiles lead the agent to discover fundamentally different intervention policies. To ground our work in contemporary policymaking challenges, we apply the model to measles outbreaks, quantifying how a modest 5% drop in vaccination coverage necessitates significantly more stringent and costly interventions to curb disease spread. This work provides a robust and adaptable framework to support transparent, evidence-based policymaking for mitigating public health crises.

## 1 Introduction

During the COVID-19 pandemic, governments faced the challenge of containing disease spread via interventions, such as lockdowns and vaccination campaigns, while mitigating economic and societal disruption. However, learning optimal intervention strategies through direct trial-and-error is neither feasible nor ethical, as suboptimal decisions carry severe consequences. Simulation-based MORL provides a framework for learning optimal strategies without risky real-world experimentation. Unlike classic RL, which collapses objectives into a single scalar reward, MORL optimizes a vector of rewards—in this case, minimizing infections and deaths while limiting socioeconomic costs (Hayes et al., 2022). This approach produces a set of Pareto-optimal policies, allowing policymakers to select strategies that best reflect their shifting priorities. The widespread availability of epidemiological statistics and government intervention data provides a fertile foundation for such simulation frameworks, while the COVID-19 crisis has underscored the importance of adaptive decision-making during public health emergencies.

However, applying RL to epidemic control presents significant challenges. Online RL requires active interaction with the environment, which is infeasible and unethical at the population level. While offline RL, which trains solely on pre-collected historical data, is an alternative, it can struggle with distributional shift and extrapolation error (Levine et al., 2020; Rifanti et al., 2025). Furthermore, a single-objective RL formulation can fail to capture the inherently multi-objective nature of pandemic response, introducing rigidity and bias (e.g., solely minimizing infections through strict lockdowns may severely harm economic activity). Instead, we leverage the available historical data to frame pandemic response as a simulator-driven MORL problem, which will allow an agent to learn and explore policies through safe, repeated trial-and-error without real-world consequences.

We develop a new SDE-based pandemic simulator which is significantly more accurate than previous state-of-the-art with comparable computational complexity, and train a PCN agent to generate non-

dominated policies balancing health and economic objectives under heterogeneous conditions. Beyond producing intervention strategies, the framework quantifies the epidemiological and economic impacts of shifting parameters such as daily contact rates, and identifies vaccination thresholds required for containment consistent with WHO recommendations. To demonstrate the framework's universality, we further extend it to diseases with distinct transmissibility and mortality profiles, and perform a case study with measles, for which vaccine hesitancy has been fueling recurring outbreaks. We quantify how insufficient vaccination coverage necessitates more stringent interventions and escalates economic losses. Together, these components provide a reproducible foundation for data-driven, priority-adaptive epidemic policymaking.

## 2 BACKGROUND AND RELATED WORK

RL offers a powerful framework for discovering adaptive intervention strategies in complex, stochastic environments, making it a good fit for epidemic control, and much prior work has employed RL to learn mitigation strategies for COVID-19 (Rifanti et al., 2025). Since learning through online, real-world experimentation is infeasible and unethical, one can either learn from historical interventions through offline learning or use a simulator to train the agent. Most applications of RL to epidemic policymaking rely on the latter, but simulator types vary in scalability and computational cost.

Agent-based models capture fine-grained interactions but are computationally expensive and difficult to scale to country-level populations (Ohi et al., 2020; Guo et al., 2022; Capobianco et al., 2021; Zong & Luo, 2022; Lin et al., 2025; Zhang et al., 2024). Compartmental SIR (Susceptible-Infected-Recovered) models and variations provide interpretable epidemic dynamics and support integration of interventions (Mai et al., 2023), though many implementations assume homogeneous mixing or simplify intervention structures, limiting realism (Wan et al., 2021; Vereshchaka & Kulkarni, 2021; Ohi et al., 2020; Libin et al., 2020). Differential equation-based simulators incorporate stochasticity that can better reflect pandemic uncertainty, but often have weak or absent validation against observed trajectories (Chadi & Mousannif, 2022; Ohi et al., 2020; Libin et al., 2020). Crucially, most existing works do not validate their simulators against empirical trajectories (Chadi & Mousannif, 2022; Desvars-Larrive et al., 2020; Guo et al., 2022; Khadilkar et al., 2020; Kompella et al., 2020; Kwak et al., 2021; Vereshchaka & Kulkarni, 2021; Mai et al., 2023; Ohi et al., 2020; Libin et al., 2020), and for the few that do, evaluation is typically confined to region-specific settings rather than global dynamics (Guo et al., 2022; Wan et al., 2021).

In addition to simulator fidelity, a review of 20 RL-based COVID-19 mitigation studies (Rifanti et al., 2025) highlights recurring challenges in terms of action space, reward design, and scalability. Many works in literature restrict the action space to overly simplified interventions, such as lockdown toggles or binary school closures, which limits the exploration of policy mixes in realistic settings (Ohi et al., 2020; Vereshchaka & Kulkarni, 2021; Chadi & Mousannif, 2022; Khadilkar et al., 2020; Kwak et al., 2021; Wan et al., 2021; Libin et al., 2020), though some recent studies have began to adopt multidimensional or continuous controls (Guo et al., 2022; Mai et al., 2023; Kompella et al., 2020). Reward design is a further bottleneck: Scalarized penalties remain common, with many studies only exploring epidemiological outcomes or collapsing multiple rewards into a single vector (Libin et al., 2020; Kwak et al., 2021; Ohi et al., 2020; Vereshchaka & Kulkarni, 2021; Chadi & Mousannif, 2022); however, some do consider more refined formulations with multi-objective rewards (Guo et al., 2022; Mai et al., 2023; Wan et al., 2021). Lastly, scalability limits applicability: Region-level models can capture granular dynamics, but such frameworks cannot be extended to other geographies (Padmanabhan et al., 2021; Chadi & Mousannif, 2022; Ohi et al., 2020; Khadilkar et al., 2020; Wan et al., 2021; Vereshchaka & Kulkarni, 2021; Guo et al., 2022; Mai et al., 2023; Kompella et al., 2020; Libin et al., 2020).

Our work addresses these gaps by developing a scalable SDE-based simulator calibrated and validated against global COVID-19 data. Furthermore, we incorporate a multi-dimensional action space and reward framework to enable a more realistic exploration of policy trade-offs.

## 3 METHODOLOGY

We frame the task of learning optimal COVID-19 interventions as a MORL problem, requiring state–action transitions that capture realistic pandemic dynamics. This includes constructing a large-scale dataset of government policies and epidemiological outcomes as well as developing a

calibrated simulator that integrates these interventions with disease-spread dynamics. This section describes the dataset, simulator design, and agent training for learning interventions strategies.

**Dataset.** The dataset combines *interventions* and *epidemiological outcomes* from 176 countries. Interventions are drawn from the *Oxford COVID-19 Government Response Tracker* (OxCGRT) (Hale et al., 2021), which records 24 policy indicators of government response (e.g., school closures and travel restrictions) across four categories: closure, economic, health, and vaccines. Epidemiological outcomes, including the daily number of new infections and COVID-19-related deaths, are sourced from *Our World in Data* (OWID) (Mathieu et al., 2020). Additionally, country-level statistics, such as landmass and population, are added from *Wikipedia* (Wikipedia, 2025a). The combined dataset provides country-level COVID-19 spread trajectories and their corresponding national interventions from 2020 to 2022. To create four distinct intervention strength indicators, we averaged the policy indicators within each of the categories: closure, economic, health, and vaccines.

Table 1: Dataset feature categories for each date (2020–2022).

| Category | Columns |
| --- | --- |
| Country | Land area, population |
| Cases | New/total cases, per million |
| Deaths | New/total deaths, per million |
| Closure Interventions | School/workplace closures, public event and transport restrictions, stay-at-home orders, internal movement limits |
| Economic Interventions | Income support, debt relief, fiscal measures |
| Health Interventions | Info campaigns, testing, contact tracing, investment in healthcare, facial coverings, vaccination policy, protection of elderly |
| Vaccine Interventions | Prioritization, availability, eligibility, support, mandatory vaccination |

**Disease Spread Simulators.** While it is possible to learn intervention policies directly from the dataset via offline learning, using the dataset for calibration and a simulator for training provides two key advantages: First, a parametric design allows the framework to be easily adapted to other pathogens by modifying the epidemiological characteristics. Second, the simulator provides a transparent and interpretable model for how interventions affect disease dynamics, enabling policymakers to translate learned strategies into interventions in their own settings. For example, closures directly affects average the number of daily contacts within the populations (see Table 2).

To ensure our simulator produces realistic national-level pandemic dynamics, we tested three types of infection-dynamics simulators from the literature: an agent-based model (Ohi et al., 2020; Guo et al., 2022; Capobianco et al., 2021; Zong & Luo, 2022; Lin et al., 2025; Zhang et al., 2024), a compartmental SIR framework (Wan et al., 2021; Vereshchaka & Kulkarni, 2021; Mai et al., 2023), and an SDE model (Xu et al., 2024). As shown in Appendix A, the agent-based and SIR simulators do not replicate the national-level pandemic dynamic nearly as well as the SDE simulator. This is not a matter of fine-tuning parameters, but of the suitability of scale—the agent-based simulator captures small-scale dynamics, but becomes too computationally costly on a national-level, whereas the SIR approach is insufficiently granular. We hence adopt the SDE-based approach and build a simulator based on the works of Xu et al. (2024) for predicting pandemic dynamics under interventions.

The infection-spread dynamics are modeled as SDE with drift terms representing deterministic progression and diffusion terms, introducing multiplicative Gaussian noise proportional to subpopulation size. This construction reflects the variability of real-world epidemics, where fluctuations in contact rates, reporting accuracy, and behavioral responses lead to both upward and downward deviations from the deterministic trend. By capturing these stochastic effects, the simulator provides a more realistic range of trajectories for training RL agents under uncertainty.

We partition the population into five groups: (i) Susceptible ($S$) are healthy individuals with no immunity to the infection; (ii) healthy/protected ($H$) are healthy with immunity to the infection due to recovery from infection or vaccination; (iii) infected ($I$) are actively spreading the infection to others; (iv) quarantined ($Q$) are currently infected, but are not spreading the infection to others; and (v) deceased ($D$) are individuals who died due to the infection. State transitions follow an epidemiological logic: $S$ decreases via infection, vaccination, or natural death; $H$ consists of vaccinated or recovered individuals with lower infection risk; $I$ represents currently infected individuals who

may recover, die, or quarantine; $Q$ contains isolated cases that no longer transmit; and $D$ records disease-induced deaths. The exact SDEs are:

$$dS = \left[ \underbrace{\omega N}_{\text{births}} - \underbrace{\sigma \mu S \mathbb{I}}_{\text{infections}} - \underbrace{(a + \beta) S}_{\text{natural death + vaccination}} \right] dt + \xi_S,$$

$$dH = \left[ \underbrace{\beta S}_{\text{vaccination}} + \underbrace{\phi I + \phi Q}_{\text{recovery}} - \underbrace{\delta \mu H \mathbb{I}}_{\text{infections}} - \underbrace{a H}_{\text{natural death}} \right] dt + \xi_H,$$

$$dI = \left[ \underbrace{\sigma \mu S \mathbb{I} + \delta \mu H \mathbb{I}}_{\text{new infections}} - \underbrace{(a + \nu + \phi + \rho) I}_{\text{death + recovery + quarantine}} \right] dt + \xi_I,$$

$$dQ = \left[ \underbrace{\rho I}_{\text{to quarantine}} - \underbrace{(a + \nu + \phi) Q}_{\text{death + recovery}} \right] dt + \xi_Q,$$

$$dD = \left[ \underbrace{\nu I + \nu Q}_{\text{disease deaths}} \right] dt + \xi_D.$$

where $\xi_\iota = w_\iota \iota dW_\iota$, $\iota \in \{S, H, I, Q, D\}$, $dW_\iota \sim N(0, dt)$, and $w$ is a diffusion coefficient.

The parameter values, their sources, and integration of interventions can be found in Table 2. There are 3 types of interventions modeled in the simulator: Vaccinations, modeled by increasing $\beta$; quarantine, modeled by increasing $\rho$; and closures, modeled by reducing the number of daily contacts, $\mu$. $\sigma$ is calibrated to the data by fitting simulated zero-intervention growth curves to corresponding curves from the data using the Kolmogorov–Smirnov test (Hoertel et al., 2020) (details in Appendix A.3.2). By modifying the COVID-19-specific parameters, such as infection rate and death, we adapt the simulator to other infections. For details on model design, see Appendix A.3.1.

Table 2: Key parameters for the SDE simulator, their sources and how interventions are integrated.

| Variable | Description | Value | Intervention | Source |
|---|---|---|---|---|
| $\omega$ | Birth rate per day | 0.000047 | 0.000047 | (The World Counts, 2025) |
| $\sigma$ | Transmission rate | 0.020 | 0.020 | Fitted |
| $a$ | Natural death rate per day | 0.000018 | 0.000018 | (Wikipedia, 2025b) |
| $\beta$ | Vaccination rate per day | 0 | $0.0005 \times a_t^{(v)}$ | — |
| $\delta$ | Transmission rate (vaccinated) | 0.005 | 0.005 | 25% of fitted rate |
| $\nu$ | Disease-induced death rate | 0.0014 | 0.0014 | (Worldometer, 2024) |
| $\rho$ | Quarantine rate per day | 0 | $0.01 \times a_t^{(q)}$ | Optimized |
| $\phi$ | Recovery rate per day | 0.14 | 0.14 | (WebMD, 2024) |
| $\mu$ | Daily interactions | 10 | $\frac{\mu}{1 + 0.2 \times a_t^{(c)}}$ | (Del Valle et al., 2007) |

Where $a_t^{(v)}$, $a_t^{(q)}$, and $a_t^{(c)}$ are action strengths for vaccinations, quarantine and closures, respectively. Sensitivity analysis for all parameter values and intervention strengths can be found in Appendix A.3.3.

**Reinforcement Learning Agent.** To learn how optimal interventions map to different strategic priorities, we consider a multi-objective setting, optimizing a reward vector corresponding to epidemiological and socioeconomic outcomes. A central challenge in MORL is approximating the Pareto front without training separate agents for each objective weighting. We use Pareto-Conditioned Networks (PCN) to overcome this issue by conditioning a single policy on a preference or desired return vector, allowing the model to produce diverse Pareto-efficient solutions from one network (Reymond et al., 2022).

The environment state is defined as the population groups $(S, H, I, Q, D)$, while the action space is three-dimensional, $A = \{0, 1, \ldots, 10\}^3$, representing discrete intervention levels for closure $a_t^{(c)}$, vaccination $a_t^{(v)}$, and quarantine $a_t^{(q)}$ policies. The reward vector is

$$\mathbf{r}_t = \left( -r_t^{(1)}, -r_t^{(2)}, -r_t^{(3)} \right) \in \mathbb{R}^3,$$

where $r_t^{(1)}, r_t^{(2)}, r_t^{(3)}$ denote new infections, new deaths, and economic impact, respectively. The latter is modeled as:

$$r_t^{(3)} = a_t^{(c)} + 5 \times a_t^{(q)} \times \frac{\mathbb{I}}{\mathbb{S}}.$$

This is because, while closures affect the whole population and cause significant economic disruption, the disruption from quarantine is proportional to the infected population—larger outbreaks force more individuals into quarantine, disrupting normal economic activity to a greater extent. In this work, we assume the impact of closure and quarantine policies to be equivalent when 20% of the population is infected, as by that point, most individuals would have come into contact with an infected person, and would be required to quarantine.

We integrate the simulator with the PCN pipeline from Felten et al. (2023) via MO-Gymnasium. A Pareto front is computed by enumerating all discrete intervention combinations over the episode horizon and is used both as a reference for training and to initialize the replay buffer. Training proceeds for episodes of 50 days (5,000 simulation steps), with minibatches of size 256. We draw the number of initially infected individuals uniformly between 1 and 20 to encourage adaptability.

**Assumptions and Limitations.** Although we strive to model the simulator as faithful to reality as possible, there are still assumptions and limitations to note. First, we do not account for partial or non-compliance with respect to interventions. However, this is a calibration issue as long as there are no significant temporal patterns. Second, we do not separate the different policies within the intervention categories (e.g., school closures from other types of closures), thus assuming equal weighting of policies within each category. Instead, we opt for a simple and interpretable model for interventions. Third, we assume that immunity, whether gained via vaccination or recovery, is permanent once acquired. This assumption is valid for short time periods only; hence, the simulator cannot be used over long periods. In this work, we focus on a 6–7 week period, for which this assumption holds. In addition, we do not model different vaccination or mortality rates based on age. With respect to rewards, the economic cost is not calibrated with real-world costs of interventions and is mostly used to estimate the relative costs of intervention strategies.

## 4 EXPERIMENTS

**Simulator fidelity to real data.** We evaluate our SDE simulator against real-world data to ensure that intervention strategies are learned under realistic epidemic dynamics. Figure 1 compares simulated and recorded new cases of COVID-19 in Italy, USA, and UK. While peaks do not perfectly align, the simulator captures the wave-like dynamics and stochastic variability typical of pandemics, as well as the overall scale and progression of infection curves. This shows the simulator is capable of reproducing national-scale dynamics over significantly different population sizes. Table 3 reports average relative AUC errors[1] across a broader set of countries, including Argentina, New Zealand, and Vietnam. We select these countries for their diversity in geography and epidemic experience, making them suitable benchmarks for validation. Compared to other simulator classes popular in this domain, our model consistently achieves the lowest errors, reproducing observed dynamics with higher fidelity. To our knowledge, this is also the first work to validate epidemic trajectories against real-world data on a global scale. Prior work rarely performs such validation (Rifanti et al., 2025; Chadi & Mousannif, 2022; Desvars-Larrive et al., 2020; Khadilkar et al., 2020; Kompella et al., 2020; Kwak et al., 2021; Vereshchaka & Kulkarni, 2021; Mai et al., 2023; Ohi et al., 2020), and the few that do (Guo et al., 2022; Wan et al., 2021) remain restricted to region-specific settings.

Table 3: Relative AUC errors between simulated and real national-level infection trajectories (10-run average). [a] Simulator modeled from Ohi et al. (2020), and [b] from Wan et al. (2021).

| Simulator | UK | US | Italy | Argentina | New Zealand | Vietnam |
|---|---|---|---|---|---|---|
| SDE | 0.2923 | 0.9109 | 0.3828 | 0.3149 | 0.7891 | 0.4926 |
| Agent-Based[a] | 251.5693 | 154.6792 | 375.0751 | 236.1093 | 2888.6116 | 413.8015 |
| SIR[b] | 479.9553 | 295.5070 | 715.1414 | 450.5191 | 5501.5458 | 788.9046 |

---

[1]Defined as the relative error with area under the curve (AUC) as the error metric: $\frac{|\text{AUC}_{\text{sim}} - \text{AUC}_{\text{obs}}|}{\text{AUC}_{\text{obs}}}$.

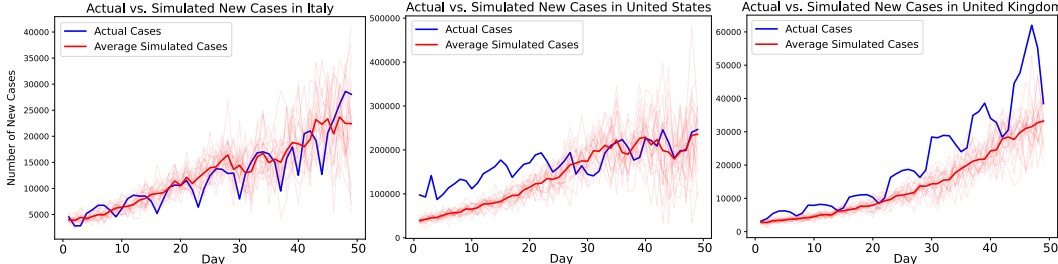

Figure 1: Simulated and observed new cases trajectories across Italy, USA, and UK. The simulator closely aligns with overall growth trends observed in the real data, demonstrating its ability to reproduce realistic epidemic progression across diverse geographies and population sizes. All 10 runs are scaled, with each simulated individual representing 1,000 people in reality.

**Adaptive Intervention Strategies.** We utilize the PCN agent's ability to learn non-dominated intervention strategies that balance three competing objectives: minimizing infections, minimizing deaths, and reducing economic disruption. By conditioning on different preference vectors and environment configurations, the agent produces tailored strategies without retraining, demonstrating robustness across heterogeneous outbreak severities and practical value for intervention strategies. All presented runs use a scaled population of 68,000 to model the UK as a baseline, with each simulated individual representing 1,000 people in reality; the parameters are extendable to other populations.

Figure 2 illustrates how the agent adapts its strategy when tasked with contrasting objectives. When balancing epidemiological and economic outcomes—defined as maintaining the minimal interventions necessary to keep infection counts below their initial level—the agent applies moderate, sustained measures that reduce infections and deaths while avoiding prolonged economic loss. Notably, the intervention strengths consistently remain below 3, coinciding with observed strengths in real-world interventions (Hale et al., 2021), suggesting that the agent learns realistic policies under non-extreme priorities. When prioritizing infection mitigation—suppressing and maintaining new cases below 10 as quickly as possible—the strategy shifts toward aggressive early interventions that proportionately ease as infections decline, reflecting adaptive adjustment to outbreak dynamics, similar to the strategy adopted by Taiwan (Chien et al., 2020). Conversely, when economic welfare is prioritized—minimizing economic loss—interventions vanish entirely, preserving activity but allowing infections and deaths to rise unchecked. While the learned policies tend to apply interventions homogeneously across action dimensions, the results clearly show that the agent can flexibly adjust to competing objectives and discover distinct trade-offs.

Figure 3 shows the approximated Pareto fronts for the strategies in Figure 2. Conditioning on a fixed horizon, 20 trajectories are collected, and the non-dominated subset forms an approximation of the front. When balancing disease spread and economic disruption, the front spans moderate trade-offs between health and economic outcomes, consistent with the compromise strategy observed earlier. Under infection mitigation, the front shifts toward minimizing cases and deaths at the expense of greater economic loss. By contrast, under economic prioritization, the front collapses to a single solution with no restrictions, preserving economic activity but incurring several multiples more infections and deaths.

**Extension to Denser Hubs.** Although many interventions are on a national scale, urban areas may require more stringent interventions to curb the spread of infection. To approximate conditions in densely populated areas, we examine how higher daily contact rates ($\mu$) affect intervention strategies and outcomes. Figure 4 illustrates trajectories for $\mu = 15$ and $\mu = 20$ compared to the $\mu = 10$ baseline. As contact rates increase, controlling disease spread requires much stronger and more prolonged interventions: Closure, vaccination, and quarantine intensities rise early to levels up to 3 times the baseline and remain elevated for most of the episode (see Appendix B.2.1). While infections are eventually suppressed, the corresponding economic cost escalates sharply, reaching more than double at $\mu = 15$ and nearly triple at $\mu = 20$. These findings suggest that in denser hubs, comparable public health outcomes demand substantially greater economic sacrifices, disproportionately burdening denser populations with limited resilience or access to remote infrastructure.

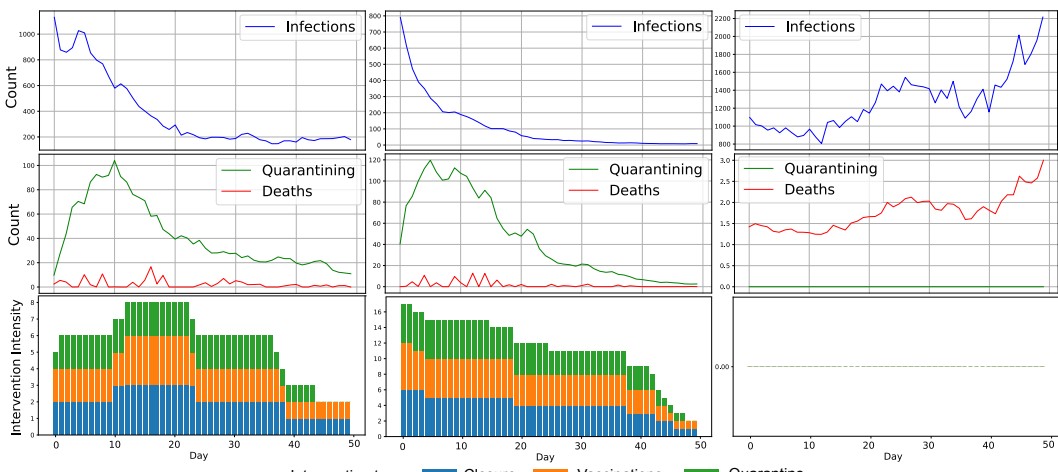

Figure 2: Number of new infections, deaths, and quarantined individuals, with corresponding interventions, throughout a 50-day episode with 1,000 initial infections ($\approx 1.5\%$ of the population) when the agent prioritizes (left) balancing disease spread and economic disruption, (middle) mitigating infections, and (right) economic welfare. On the left, the agent applies moderate, sustained measures to control spread without severely restricting economic activity. In the middle, stringent, prolonged interventions curb the spread. On the right, no interventions are applied, demonstrating an extreme prioritization of economic activity, resulting in a sharp increase in new infections.

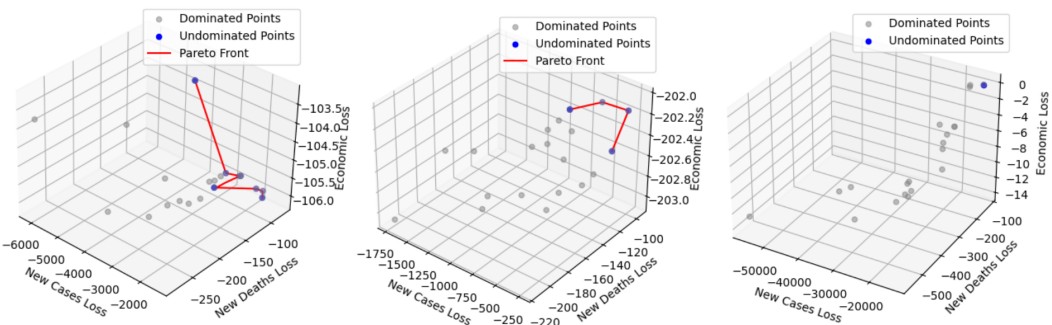

Figure 3: Pareto fronts with 1,000 initial infections when the agent prioritizes (left) balancing disease spread and economic disruption, (middle) mitigating infections, and (right) economic welfare. Corresponding to Figure 2, the approximated Pareto fronts show the optimal trade-offs between new infections, new deaths, and economic costs when prioritizing each intervention strategy.

Denser hubs also suffer more severe epidemiological consequences when constrained by the same economic budget. With equalized economic cost across scenarios, infection peaks rise to 4 times the baseline at $\mu = 15$ and nearly 16 times at $\mu = 20$, while death peaks reach 3 and 15 times higher, respectively. Quarantining levels also surge to 2 and 9 times the baseline, respectively, indicating greater disruption to daily life. Under these conditions, denser hubs experience more intense epidemiological repercussions despite an equivalent economic burden. This also implies that healthcare systems in such areas would reach full capacity much sooner, increasing the risk of overwhelming hospitals and reducing the quality of care. Detailed results can be found in Appendix B.2.2.

**Extension to Other Diseases.** To evaluate adaptability beyond COVID-19, we reparameterize the simulator to reflect the epidemiological characteristics—namely transmissibility and mortality—of polio and influenza (for more details, see Appendix B.3.1). Figure 5 illustrates example intervention strategies under the same initial conditions and objectives as the COVID-19 baseline .

With higher transmissibility and lethality, polio demands substantially stronger interventions. Under the same initial conditions and objectives, the economic impact in Figure 5 are 3 times stronger

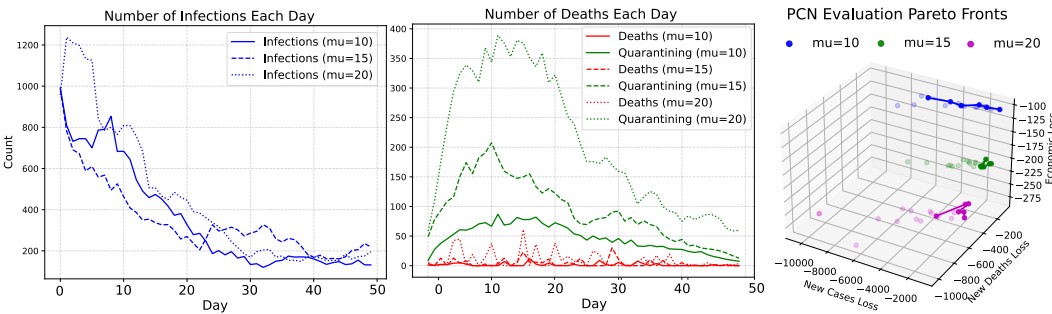

Figure 4: Comparison of intervention strategies and outcomes under higher daily contact rates of $\mu = 15$ and $\mu = 20$. As contact rate increases, interventions must become substantially more stringent and prolonged to contain disease spread (see Appendix B.2.1), leading to disproportionately higher economic burdens.

than those for COVID-19 (Figure 2a). Despite broadly similar infection trajectories, peak deaths and quarantining individuals are 4 times higher. In contrast, influenza simulations exhibit much milder dynamics, requiring only minimal interventions: Mild quarantining dominates the strategy, while closure and vaccination policies remain negligible throughout most of the trajectory (see Appendix B.3.2), reflecting real-world practice around combating seasonal flu. Infection trajectories are also broadly similar, but the economic impact is reduced to about half, peak deaths are 60% of COVID-19's, and peak quarantining levels are about 70%.

Together, these results demonstrate that the framework can be flexibly adapted to different pathogens by tuning epidemiological parameters, with disease-specific characteristics fundamentally reshaping the trade-off landscape. Crucially, this enables the quantification of how alternative disease profiles translate into different epidemiological and economic burdens, allowing policymakers to anticipate and prepare for the distinct challenges posed by future outbreaks.

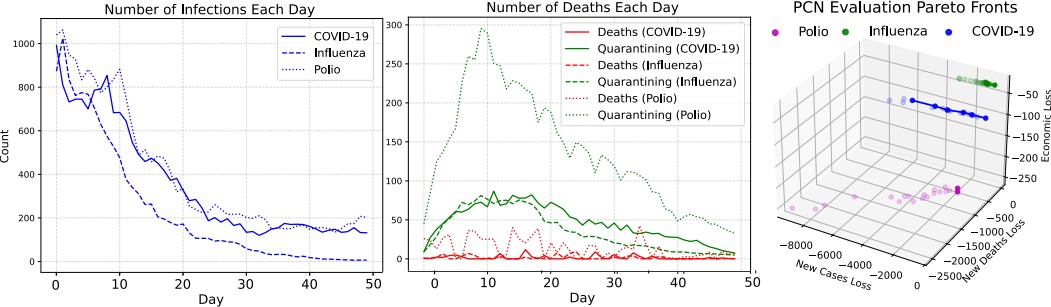

Figure 5: Comparison of intervention strategies for polio and influenza under the same initial conditions and objectives as the COVID-19 baseline. Polio requires far stronger interventions (see Appendix B.3.2) and still incurs higher health losses, while influenza can be managed with mild measures and minimal economic disruption.

**Extension to Vaccination Coverage.** Declining rates of childhood vaccinations in certain areas have led to an increase in measles outbreaks. To investigate local outbreaks, we reparameterize the simulator to measles epidemiology (see Appendix B.3.1 for full details) and initialize a small community of 1,000 individuals with a single initial case, reflecting the scenario of an outbreak beginning in an under-vaccinated community. Figure 6 compares epidemic trajectories under population vaccination rates of 95%, 90%, 85%, and 80%. At 95%, the WHO's recommended threshold (Centers for Disease Control and Prevention (CDC), 2025), transmission is effectively suppressed with no need for additional interventions. At 90% and 85%, corresponding to realistic levels in the UK (BBC News, 2025a;b), outbreaks become increasingly difficult to contain, with the 85% case experiencing infections at persistent levels. At 80%, infections already rise monotonically. Once coverage falls to 85% or below, additional measures are necessary to prevent persistent spread. Figure 6 illustrates

the minimal intervention trajectory to ensure a steady decline in new infections (detailed epidemiological and economic outcomes shown in Appendix B.4). Notably, reducing coverage from 85% to 80% results in more stringent and persistent interventions with triple the associated economic losses. These results mirror expert guidance, reinforcing the WHO's 95% recommendation, and illustrate varying degrees of disease spread below this level. By visualizing and quantifying the impact of declining vaccination coverage, the framework can support policymakers in public-health decisions.

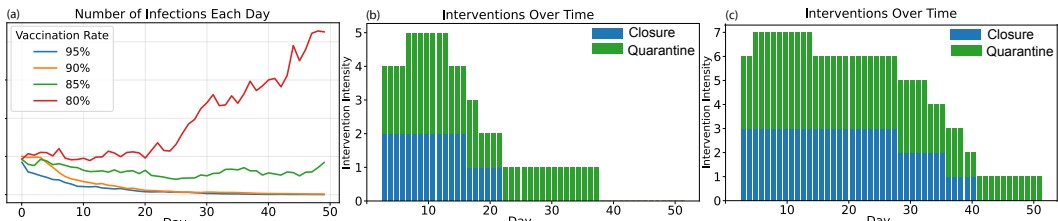

Figure 6: Epidemic trajectories and intervention strategies for measles under varying vaccination rates. The left panel shows daily new infections (full details in Appendix B.4). The middle and right panels illustrate minimal interventions necessary to control spread at 85% and 80%, where outbreaks persist. Vaccination actions are excluded to reflect fixed coverage in vaccine-hesitant communities.

## 5   DISCUSSION AND CONCLUSION

We explored the potential of MORL for epidemic intervention planning, demonstrating that an agent can learn to navigate the complex trade-offs between public health and economic stability. Our results show that the agent adapts its strategies across various policy priorities and disease characteristics. However, the framework's utility is best understood not as a tool for predicting exact case numbers, but for quantifying the relative consequences of policy choices. For decision-makers, it can offer a dynamic "what-if" engine to explore critical strategic questions: What is the likely cost of delaying interventions? If we wait, will we need measures that are twice as stringent, and will the economic impact be three times greater? By making these trade-offs explicit, the framework can provide overarching strategic guidance that is valuable even when absolute values require further calibration.

A key feature of our approach is the transparent and interpretable design of the simulator (e.g., the intensity of "closure" actions directly scales the number of daily contacts). This simplicity is intentional, as it allows policymakers to more easily translate the model's abstract intervention levels into concrete, real-world policies by estimating their impact on population-level contact rates, which may vary nationally or even locally. This interpretability also lends credibility to the model's uncalibrated adaptations. For instance, while direct trajectory data for polio and influenza was not used for validation, the framework demonstrates a strong qualitative validation. The model correctly learns that influenza can be managed with mild interventions like vaccination, whereas polio requires highly aggressive measures in the absence of widespread high compliance with vaccination recommendations. Notably, in the measles case study, the model independently corroborates WHO's vaccination guidance, identifying the tipping point where additional interventions become unnecessary. This emergent alignment supports the realistic nature of the model's underlying epidemiological logic.

Beyond strategic planning, the framework may be useful as a communication tool to explain and support policy decisions. By visualizing the direct link between community actions (such as vaccination uptake) and the necessity of restrictive measures, this tool may help make public health guidance more transparent and accessible.

Our work highlights the promise of simulation-based MORL as a decision-support tool in public health. By integrating a realistic and interpretable simulator with a multi-objective agent, we offer a blueprint for how machine learning can generate, explore, and communicate a diverse set of adaptable intervention strategies. Our findings suggest that such frameworks have the potential to complement expert judgment and enhance data-driven decision-making in future public health crises.

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

# A    CONSIDERED SIMULATORS

## A.1    AGENT-BASED SIMULATOR

This discarded simulator follows the work of Ohi et al. (2020), modeling a population of Person objects that move randomly on a grid and transmit infection upon contact after an incubation period.

**Methodology.**    Each Person progresses through infection, recovery, or death according to predefined COVID-19 statistics (e.g., infection rate, mortality rate, incubation period). Interventions directly alter movement and infectiousness: Closures restrict mobility, mask and distancing measures reduce transmissibility, and vaccination modifies susceptibility and mortality risk. Economic welfare is tied to population mobility. Assumptions include uniform spatial distribution, no reproduction, and perfect compliance with linear intervention effects.

**Experiments.**    While conceptually illustrative, the simulator is computationally infeasible at a country scale. Each Person must be tracked individually, and for the UK alone, initialization at full scale would require several hours (Figure 7). Running on smaller samples circumvents runtime issues but quickly saturates, where disease spread exhausts the limited population, causing new cases to collapse to zero (Figure 8). Scaling results back up fails to reproduce realistic epidemic dynamics.

**Conclusion.**    Agent-based simulation cannot feasibly capture country-level pandemic trajectories: It is intractable at full populations and inaccurate at reduced scales. As such, it is unsuitable for training RL agents requiring several accurate, country-level simulation runs.

## A.2    GENERALIZED SIR SIMULATOR

This discarded simulator is based on the work of Wan et al. (2021), which extends the classic SIR framework into a GSIR (generalized SIR) model. In GSIR models, stochasticity is introduced by sampling new infections from a Poisson distribution, and recoveries from a Binomial distribution, which are popular choices in literature (Qu et al., 2022). Formally, the transition model can be defined through the following equations:

$$X_{I,t+1}^S = X_{I,t}^S - e_{I,t}^S, \quad e_{I,t}^S \sim \text{Poisson}\Big(\sum_{j=1}^{J} \beta_j \, I_{j,t}\big(A_{I,t}=j\big) \, \frac{X_{I,t}^S}{M_I}\Big)$$

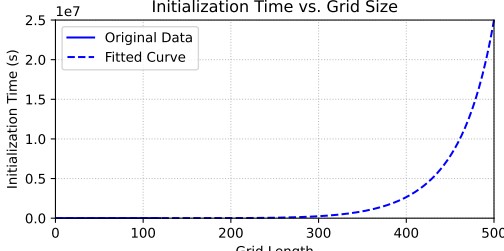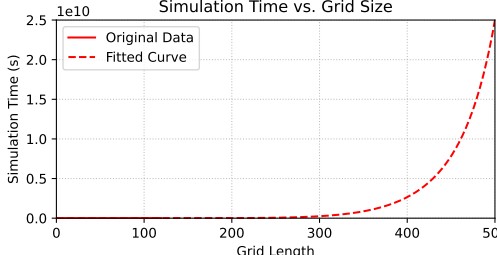

Figure 7: Initialization and simulation runtimes for the agent-based simulator at varying grid lengths with the same population density as the UK. Measurements are taken for grid lengths from 10 to 100 in increments of 10 and then extrapolated to a length of 500, which corresponds to the UK's size in this configuration. At this scale, initialization alone would require several hours, and the full simulation substantially longer. Such runtimes are infeasible for repeated training runs, especially given that the UK is a relatively smaller country in the dataset.

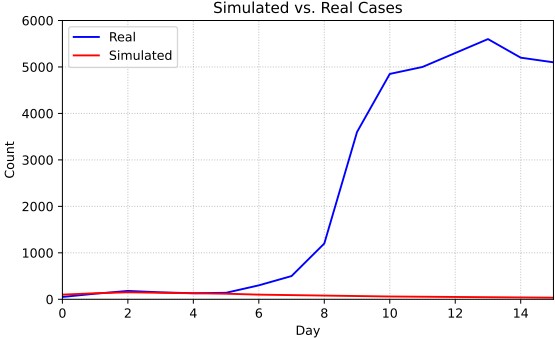

Figure 8: Simulated and real numbers of new cases across 10 runs using the agent-based simulator. When scaled down to a smaller sample of the population for computational feasibility, the simulated trajectories collapse after a few days due to the entire simulated population becoming infected too quickly. This loss of range renders it infeasible to reproduce realistic epidemic patterns at a country-level scale, limiting the simulator's suitability for the RL framework.

$$X_{I,t+1}^R = X_{I,t}^R + e_{I,t}^R, \quad e_{I,t}^R \sim \text{Binomial}\big(X_{I,t}^I, \zeta\big)$$

$$X_{I,t+1}^L = M_I - X_{I,t+1}^S - X_{I,t+1}^R,$$

where:

- the superscript of $X_{l,t}$ denotes the number of susceptible $S$, infectious $I$, or removed $R$ (isolated, recovered, or deceased) individuals in region $l$ at time $t$;

- $M_l$ denotes the total population of region $l$;

- $A_{l,t} \in \{1, \ldots, J\}$ denotes the discrete intervention level applied in region $l$ at time $t$;

- $\beta_j$ is the per-contact infection rate under intervention level $j$;

- $\zeta$ is the removal probability per infectious individual per time step;

- and the superscript of $e_{l,t}$ represents the number of new infections $S$ or removals $R$.

**Methodology.** Interventions are collapsed into a single discrete action $A \in \{1, \ldots, J\}$, with all 24 OxCGRT policy indicators normalized and uniformly averaged into one control variable. This simplification assumes equal weighting across heterogeneous policies (e.g., school closures vs. travel bans), and alignment between OxCGRT (Hale et al., 2021) stringency levels and the original dataset, which focuses solely on China.

**Experiments.** This approach avoids the scalability limits of the agent-based model but introduces its own drawbacks. Collapsing 24 policies into a single action prevents differentiation between distinct interventions, limiting policy realism. Moreover, simulated trajectories consistently exhibit a single infection peak that collapses to zero (Figure 9). Even with stochastic sampling, the curves remain overly smooth and fail to reproduce the wavelike fluctuations typical of real pandemic data.

**Conclusion.** Although computationally efficient, the generalized SIR simulator oversimplifies intervention dynamics and lacks fidelity to observed epidemic variability. Its inability to capture short-term fluctuations or the effects of tightening and loosening policies renders it unsuitable for training RL agents to learn realistic intervention strategies.

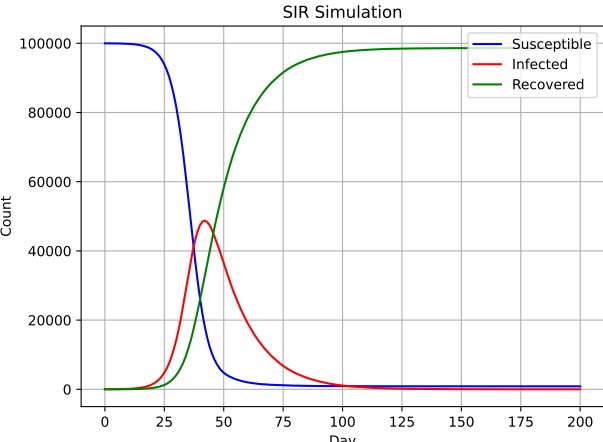

Figure 9: Example simulation run using the generalized SIR simulator, displaying the progression of susceptible, infected, and recovered populations over time. Despite the introduction of stochasticity through random sampling for infections and recoveries, the resulting curves remain overly smooth and monotonic, which does not reflect fluctuations in real-world pandemic dynamics.

## A.3 SDE SIMULATOR

Unlike the discarded agent-based and GSIR simulators, the SDE model produces irregular, wavelike fluctuations in new cases and deaths (Figure 10), reflecting the stochastic variability observed in real-world pandemic data. This higher fidelity makes it substantially more suitable for capturing epidemic dynamics and for training RL agents under uncertainty.

### A.3.1 MODEL DESIGN

This part delves into the model design of the proposed SDE simulator. The state transitions follow epidemiological logic for a compartmental COVID-19 model, and the transitions between states, as well as corresponding parameters, are depicted in Figure 11. The dynamics of each state can be effectively summarized:

- The susceptible group ($S$) comprises individuals unprotected from COVID-19. This group increases through births at a daily global birth rate $\omega$, and decreases via infection at rate $\sigma$, vaccination at rate $\beta$, and natural death at rate $a$. Individuals who are vaccinated or recover from infection join the healthy/protected group ($H$) and cannot reenter $S$, as their infection rate differs from that of unprotected susceptibles. Although the protection from vaccines are temporary in reality (Centers for Disease Control and Prevention, 2025b), the protection duration exceeds that of the simulation period in this investigation.
- The healthy/protected group ($H$) consists of individuals resistant to COVID-19, either through vaccination or recovery from infection. This group grows via the vaccination of susceptibles ($\beta$) and recovery of infected individuals ($\phi$). It decreases through infection at rate $\delta$ (which is lower than $\sigma$) and natural death ($a$).

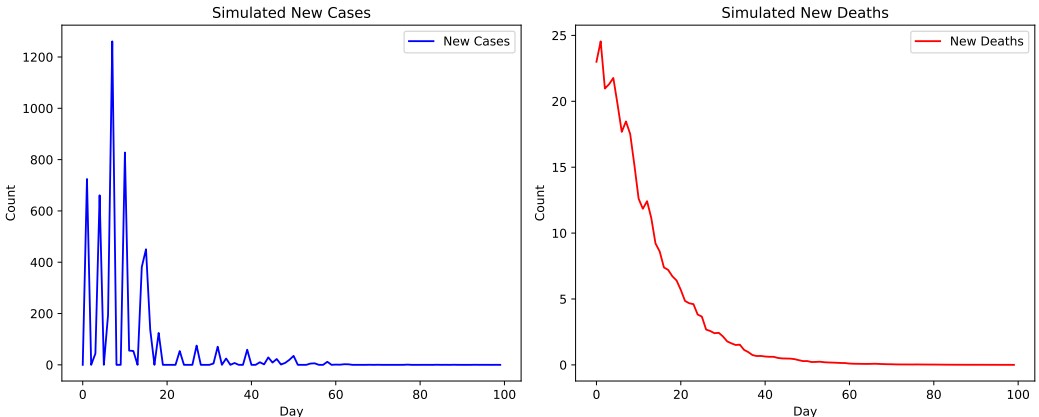

Figure 10: Example simulation run using the SDE simulator, visualizing daily new cases and deaths. The model produces irregular fluctuations in its trajectories due to the introduction of stochastic noise, which more closely resembles real-world pandemic patterns compared to the other simulators.

- The infected group ($I$) contains individuals currently infected with COVID-19. Members enter from infections in $S$ or $H$, and exit through recovery ($\phi$), natural death ($a$), disease-induced death ($\nu$), or transfer to quarantine ($\rho$).

- The quarantined group ($Q$) is a subset of $I$ who does not transmit the virus. This group increases when infected individuals quarantine ($\rho$) and decreases through recovery ($\phi$) or death ($\nu$). As a modeling simplification, $Q$ does not distinguish between vaccinated and unvaccinated individuals; all share the same recovery and death rates.

- The deceased group ($D$) includes individuals who have died from COVID-19. Members enter from deaths in $I$ and $Q$ at rate $\nu$ and cannot leave this group.

- The infection rates $\sigma$ and $\delta$ for susceptibles and protected individuals, respectively, depend on the number of daily interactions ($\mu$) and proportion of the population infected and not quarantining ($\mathbb{I}$). The product of these values represents the expected number of interactions with an infected individual, which can then be multiplied by the corresponding transmission rate to determine transition rates from these groups into $I$.

The use of SDE is justified by the inherent variability in epidemic dynamics. Real-world disease spread is subject to random fluctuations in contact rates, reporting accuracy, and individual responses to interventions. The drift terms in the model from Section 3 hence capture the average progression of each compartment, whereas the diffusion terms $\xi$ introduce multiplicative noise proportional to compartment size, representing proportional variability in transitions. Finally, the Wiener increments $dW$ are zero-mean Gaussian variables, allowing both upward and downward deviations from the deterministic trend. This formulation yields a more realistic and flexible model that can capture the range of possible epidemic trajectories, allowing the training of RL agents under uncertainty.

### A.3.2 PARAMETER SEARCH

**Methodology.** A critical parameter in the SDE simulator is the transmission rate $\sigma$, which governs spread among unprotected individuals in the absence of interventions. Unlike other parameters that can be directly drawn from literature (e.g., recovery $\phi$, mortality $\nu$), $\sigma$ must be empirically calibrated. To accomplish this, we extract exponential growth curves from periods of uncontrolled spread (no interventions) in the dataset. The simulator is then run under equivalent zero-intervention conditions for a range of candidate $\sigma$ values (0.010–0.030), and the resulting growth distributions (Figure 12) are compared to observed data using the Kolmogorov–Smirnov (K-S) test (Hoertel et al., 2020). The value that best matches empirical distributions is selected for subsequent experiments.

**Experiments.** Sensitivity analysis indicates that $\sigma = 0.020$ best reproduces observed epidemic dynamics, with a maximum CDF difference of 21.7% and $p = 0.066$, meaning the null hypothesis

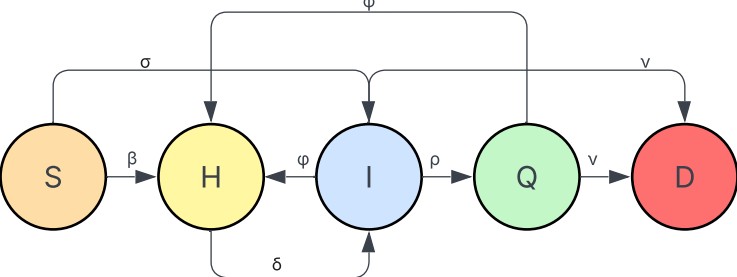

Figure 11: Schematic representation of the SDE model, illustrating the transitions between susceptible (S), healthy/protected (H), infected (I), quarantined (Q), and deceased (D) groups. Births ($\omega$) add to the susceptible population, while all compartments experience natural mortality ($a$).

of similarity cannot be rejected at the 5% level (Table 4). This calibration is performed across multiple geographies, ensuring robustness to diverse contexts. Example runs (Figures 10) demonstrate that the model captures the stochastic peaks and troughs typical of pandemic dynamics, unlike the discarded simulators. Further results, including country-level trajectory comparisons, are presented in Figure 1.

Table 4: K-S test results for several stochastically simulated infection growth rate distributions, each generated using a different $\sigma$ value in the SDE simulator, compared to the real distribution from recorded countries.

| $\sigma$ | K-S Statistic ↓ | p-Value ↑ | $\sigma$ | K-S Statistic ↓ | p-Value ↑ |
|---|---|---|---|---|---|
| 0.010 | 0.940 | 0.000 | 0.020 | 0.217 | 0.066 |
| 0.011 | 0.881 | 0.000 | 0.021 | 0.229 | 0.022 |
| 0.012 | 0.868 | 0.000 | 0.022 | 0.420 | 0.000 |
| 0.013 | 0.790 | 0.000 | 0.023 | 0.523 | 0.000 |
| 0.014 | 0.733 | 0.000 | 0.024 | 0.564 | 0.000 |
| 0.015 | 0.694 | 0.000 | 0.025 | 0.687 | 0.000 |
| 0.016 | 0.616 | 0.000 | 0.026 | 0.705 | 0.000 |
| 0.017 | 0.502 | 0.000 | 0.027 | 0.817 | 0.000 |
| 0.018 | 0.354 | 0.000 | 0.028 | 0.848 | 0.000 |
| 0.019 | 0.254 | 0.008 | 0.029 | 0.901 | 0.000 |

**Conclusion.** Calibrating $\sigma$ ensures the simulator achieves high fidelity to real-world growth trends, producing wavelike dynamics across countries that closely match observed epidemic patterns. This makes it a suitable basis for training RL agents to explore intervention strategies under realistic uncertainty.

### A.3.3 SENSITIVITY ANALYSIS

To assess the sensitivity of the SDE simulator to parameters that can realistically vary (e.g., contact rate, quarantine rate, recovery rate) as well as intervention strengths, we systematically tested a range of values for each, while keeping constants such as natural birth and death rates, the COVID-19 death rate, and the transmission rate ($\sigma$ already calibrated against real data) fixed. We ran each configuration 10 times, and the highlighted values in Table 5 denote the best-performing values selected for the final simulator. Parameters were varied symmetrically around their baseline to illustrate how simulator accuracy changes when values are increased or decreased, except where the baseline was already zero. As expected, the transmission rate $\mu$ had the strongest effect: Even small deviations from the baseline caused dramatic reductions in fidelity. Vaccination rate $\beta$ also

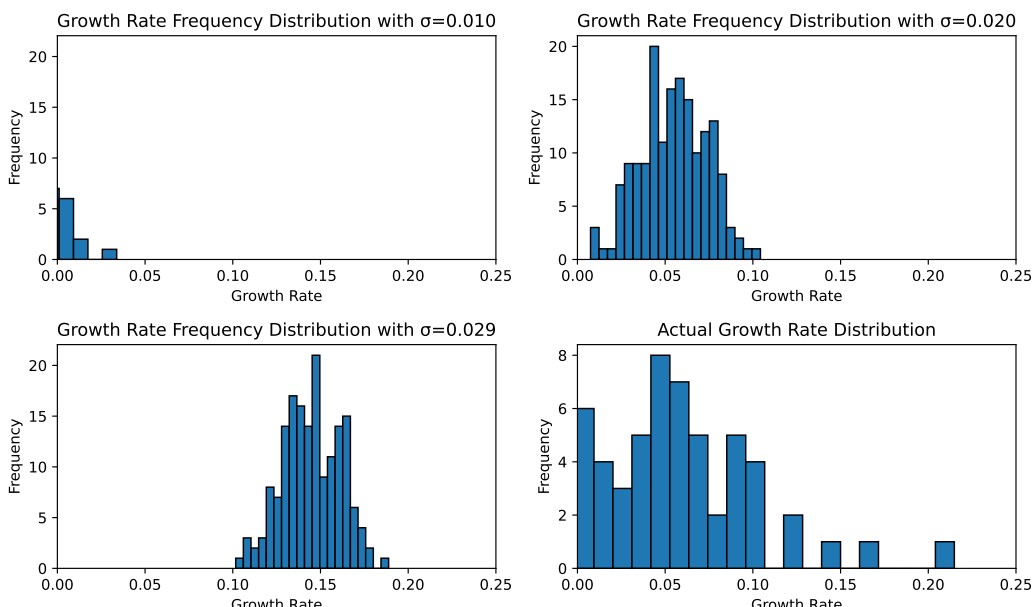

Figure 12: Growth rate distributions of infection counts at $\sigma = 0.010, 0.020, 0.029$ compared to that of the true data across the recorded countries. These comparisons are part of the parameter calibration process used to align the simulator with observed COVID-19 growth trends under zero-intervention conditions. Among the tested values, $\sigma = 0.020$ produces a distribution most similar to the real data distribution, supporting its selection for the simulator in subsequent experiments.

proved highly sensitive, with the best value aligning at zero in pre-intervention settings, while the recovery rate $\phi$ performed best at seven days, consistent with COVID-19 clinical patterns. Other parameters showed relatively minor differences, though increments were chosen conservatively to reflect realistic scales relative to the selected value.

In addition to parameter values, we also conducted sensitivity analysis on intervention strengths. Table 6 delineates the relative AUC errors when closure, vaccination, and quarantine measures are applied at varying intensities. Results indicate that closure policies exert the strongest influence on simulator fidelity, with higher stringencies resulting in higher deviations in error. Quarantine measures also have a substantial effect, while vaccination strengths produce relatively smaller changes in error. This ranking is intuitive: Closures and quarantine directly reduce contact rates and transmission opportunities, whereas vaccination acts more indirectly, with delayed effects.

| Parameters | Tested Values | Relative AUC Error | | |
|:---:|:---:|:---:|:---:|:---:|
| $\mu$ | $\{9, \mathbf{10}, 11\}$ | 0.6384 | 0.3934 | 1.4702 |
| $\beta$ | $\{\mathbf{0}, 0.01, 0.1\}$ | 0.3934 | 0.6621 | 0.9783 |
| $\delta$ | $\{0.001, \mathbf{0.005}, 0.01\}$ | 0.3938 | 0.3934 | 0.4087 |
| $\phi$ | $\{0.1, \mathbf{0.14}, 0.2\}$ | 5.7529 | 0.3934 | 0.9913 |
| $\rho$ | $\{0.01, \mathbf{0.05}, 0.1\}$ | 0.4146 | 0.3934 | 0.4690 |

Table 5: Tested parameter values for the SDE simulator and the corresponding average relative AUC errors, used to assess the model's sensitivity to variations in each parameter. The highlighted/bolded values correspond to those selected for the final simulator.

| | **Intervention Strength** | | |
|---|---|---|---|
| **Interventions** | **0** | **1** | **3** |
| **Closure** | 0.3934 | 0.7982 | 0.9994 |
| **Vaccine** | 0.3934 | 0.3511 | 0.4125 |
| **Quarantine** | 0.3934 | 0.4847 | 0.8243 |

Table 6: Tested intervention strengths for the SDE simulator and the corresponding average relative AUC errors, used to assess the model's sensitivity to variations in each intervention.

## B  REINFORCEMENT LEARNING AGENT

### B.1  OUTBREAK SEVERITIES

Figure 13 presents intervention strategies and epidemic trajectories under different initial infection levels (10, 100, and 1,000 cases).

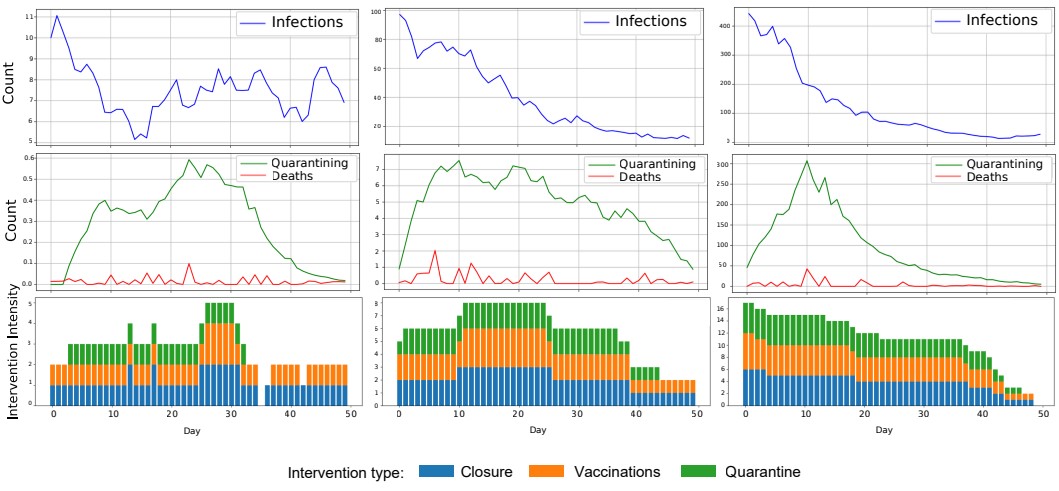

Figure 13: Agent's strategy to mitigate the number of new infections and deaths throughout an episode given 10 (left), 100 (middle), and 1,000 (right) initial infections. With low initial infections, the agent applies loose, sustained interventions to prevent escalation, keeping new infections and deaths minimal. For higher initial infections, particularly in the right panel, the agent implements correspondingly stronger early interventions to quickly suppress the outbreak before a gradual relaxation as cases decline. These results demonstrate the agent's ability to adapt its control policy to different outbreak severities while balancing intervention intensity with infection suppression.

### B.2  CONTACT RATES

#### B.2.1  INTERVENTIONS FOR DISEASE CONTROL

The corresponding intervention strategies for higher contact rates are presented in Figure 14, illustrating how policies evolve under $\mu = 15$ and $\mu = 20$.

#### B.2.2  FIXED ECONOMIC IMPACT

Figure 15 presents epidemic trajectories under increased daily contact rates ($\mu = 15, 20$) when interventions are constrained to the same overall economic cost as in the baseline.

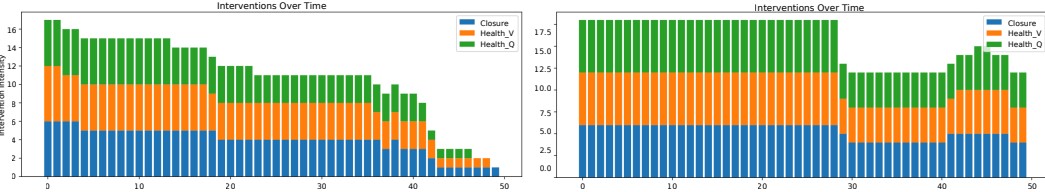

Figure 14: Intervention trajectories required to contain disease spread under higher daily contact rates. The left panel shows $\mu = 15$, where moderate but sustained interventions are necessary, while the right panel shows $\mu = 20$, where interventions must remain stringent for most of the episode. These results illustrate how higher contact rates substantially increase both the duration and intensity of required interventions.

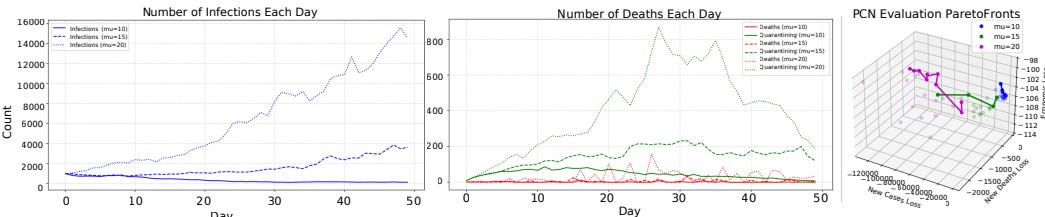

Figure 15: Example intervention strategy with 1,000 initial cases and elevated contact rates of $\mu = 15$ and $\mu = 20$, each constrained with same economic loss as the baseline in Figure 3a. Notably, increasing contact rates leads to substantially higher peaks in new infections and deaths despite equal economic cost. The corresponding Pareto fronts confirm regions with significantly worse epidemiological outcomes for the same economic input, highlighting that denser hubs are disproportionately impacted.

## B.3  OTHER DISEASES

### B.3.1  SIMULATOR REPARAMETERIZATION

The simulator is reparameterized to reflect the epidemiological characteristics of polio and influenza in Section 4.

**Polio.** The transmission rate is set to 1.75 times that of COVID-19 (McCandless et al., 2018), the mortality rate to 23% (McCandless et al., 2018), and the daily recovery rate to 0.1 (World Health Organization, 2025). Daily case reporting is unavailable, and intervention records such as vaccination campaigns, quarantines, and travel restrictions are inconsistently documented, precluding direct validation of simulated outputs against real-world data.

**Influenza.** The simulator parameters are set to half the COVID-19 transmission rate (McCandless et al., 2018), a mortality rate of 0.1% (McCandless et al., 2018), a daily recovery rate of 0.14 (Villines, 2025), and vaccination efficacy of approximately 50% (Centers for Disease Control and Prevention, 2025a). As comprehensive daily case and intervention datasets are limited and under-reported, only parameter adjustments were applied, without further calibration.

**Measles.** To reflect the measles disease profile, the transmission rate is set to 4.5 times that of COVID-19 (McCandless et al., 2018), a mortality rate of 0.8% (McCandless et al., 2018), daily recovery rate of 0.12 (NHS, 2025), and vaccination efficacy of 99% (NHS, 2024). Since infection and intervention data are likewise limited, only parameter adjustments were applied.

### B.3.2 INTERVENTIONS FOR POLIO AND INFLUENZA

The corresponding intervention strategies for polio and influenza are shown in Figure 16, highlighting how the agent adapts to different epidemiological profiles under the same initial conditions and objectives as the COVID-19 baseline.

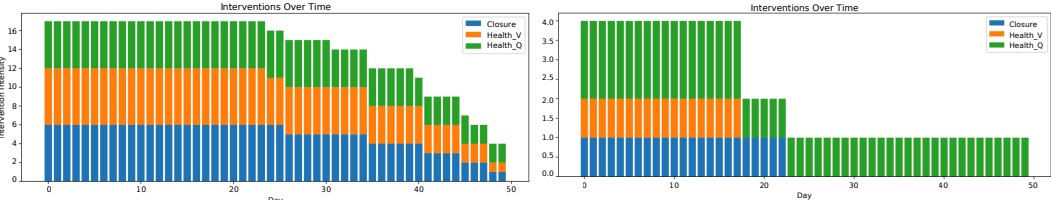

Figure 16: Comparison of intervention strategies for polio (left) and influenza (right) under the same initial conditions and objectives as the COVID-19 baseline. Polio requires substantially stronger and more prolonged interventions, whereas influenza can be contained with mild measures and minimal economic disruption.

### B.4 VARYING VACCINATION RATES

Figures 17 and 18 present example intervention strategies and epidemic trajectories when the population has a fixed vaccination rate of 85% and 80%, respectively.

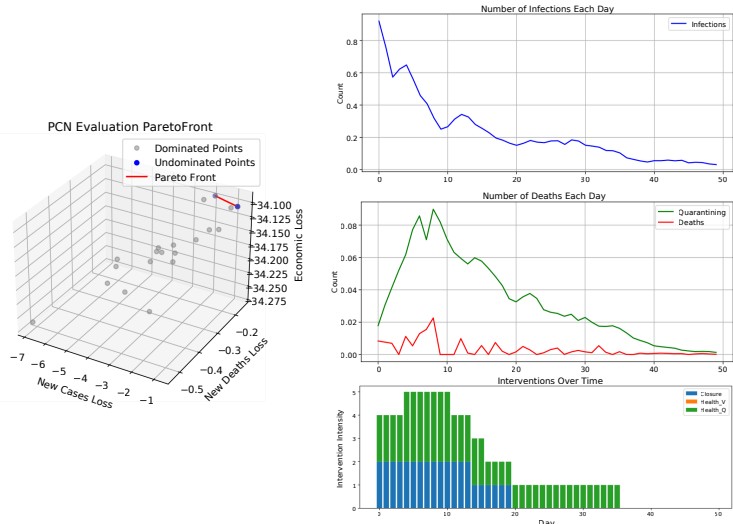

Figure 17: Comparison of intervention strategies for measles with an 85% vaccination rate under the same initial conditions as Figure 6. Mild closure and quarantine interventions are required to contain disease spread.

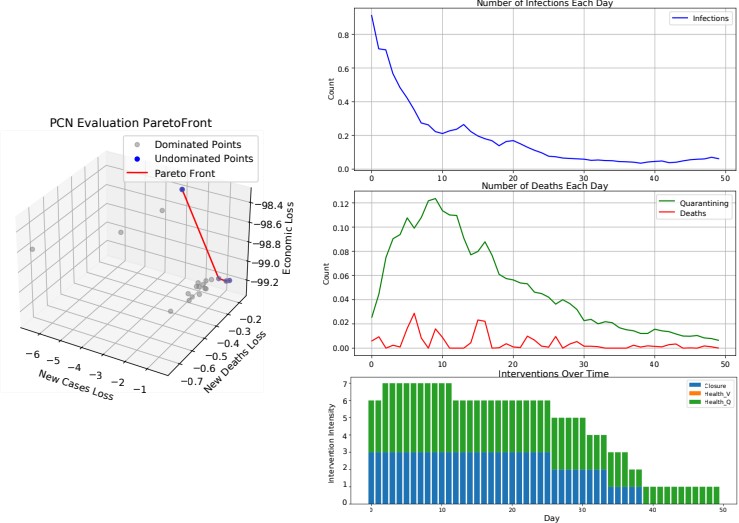

Figure 18: Comparison of intervention strategies for measles with an 80% vaccination rate under the same initial conditions as Figure 6. Comparatively stronger and persistent closure and quarantine interventions are required to contain disease spread.

