# OpenReview forum: "Learning Pareto-Optimal Pandemic Intervention Policies with MORL"
_ICLR.cc/2026/Conference — Submitted to ICLR 2026_

### Official Review · Reviewer_co3W · 2025-10-30

**Soundness:** 1
**Presentation:** 2
**Contribution:** 1
**Rating:** 2
**Confidence:** 4

**Summary:**

This paper addresses the modeling and evaluation of pandemic intervention policies through multi-objective reinforcement learning (MORL), proposing a framework that balances disease containment with socioeconomic stability. The authors develop a stochastic differential equation (SDE) based pandemic simulator calibrated against global COVID-19 data from 176 countries and employ a Pareto-Conditioned Network (PCN) to learn diverse intervention policies. The framework is validated on COVID-19 data and extended to other infectious diseases (polio, influenza, measles) to demonstrate generalizability. The claimed contribution is providing a scalable, data-driven approach to multi-objective pandemic policy optimization with improved simulator fidelity compared to prior work.

**Strengths:**

Applying reinforcement learning to pandemic intervention design is a timely and high-impact application area. The use of multi-objective reinforcement learning (MORL) with Pareto-Conditioned Networks (PCNs) to capture trade-offs between health and economic objectives is well motivated and technically sound. Modeling interventions at the national scale is realistic and relevant for large-scale policy-level decision support. The paper is clearly written and easy to follow.

**Weaknesses:**

## Major weaknesses
### Contribution/Novelty
Overall, it is not clear what the contributions of this work are. The authors should consider adding bullet points at the end of the introduction to summarize the main contributions of this work.
* The authors claim “Our simulator reproduces national-scale pandemic dynamics with orders of magnitude higher fidelity than other models commonly used in reinforcement learning (RL) approaches to pandemic intervention”. However, the simulator has the exact same formulation as the one by [1], who in turn followed the one by [2]. Although the authors claim their simulator is based on the one by [1] who also used it to model the effect of diverse transmission rates of Covid-19, the differences with respect to such simulators are not clearly specified. The only difference seems to be the inclusion of the “deceased” group of individuals. Hence, the novelty with respect to that work seems limited. The authors should clearly state what in the mathematical model in lines 165-178 is taken from [1,2], and should clearly highlight any introduced novelty. Also, they should compare their simulator against the one by [2] to actually motivate the introduced changes.
* [3] also used MORL to find a set of Pareto-optimal policies. Although this work is cited in the related work, the authors should clearly highlight the differences with the approach by [3], and compare their results with the ones by [3].

### Soundness
* For several experiments, it is not clear how many runs were performed and whether the plots actually show a mean over multiple runs. In general, no standard deviations across runs are reported in any of the presented results (even the ones where the presence of multiple runs is specified).
* The AUC error in Table 3 is lower for the UK than for Italy and the US, but that does not seem to be the same from Figure 1. The authors should clarify this in the discussion.
* The authors claim “Our simulator reproduces national-scale pandemic dynamics with orders of magnitude higher fidelity than other models commonly used in reinforcement learning (RL) approaches to pandemic intervention” is not convincingly supported by experimental evidence. The qualitative evaluations of the competing methods in Appendix A seems oversimplistic. For instance, results by [3] with a generalized SIR simulator (Figure 4) do not show single infection peaks. What changes were introduced with respect to [3] that might cause this significant degradation? The authors should clearly state how the simulators were implemented.
* The reward computation modeling the economic impact seems rather arbitrary. The authors should discuss the design of such reward further, discuss the selection of the 5x multiplier and the sensitivity of the loss to it. This weakens any conclusion on the trade-off between epidemiological control and the economic stability, which is central to the work.
* The authors include just results for data from the UK in the analyses in pages 6 and 7, and do not report results for other countries in the appendix.
* Claims of adaptation to other diseases are qualitative only and not validated on real data. The authors should either soften the generalizability claim, or add validation on real data.

### Reproducibility
The authors should include a table in the appendix with all the hyperparameters needed to reproduce their results, including the ones used to train the PCNs. Results are not currently easily reproducible.

## Minor weaknesses
### Clarity
The paper is overall well written, but there are a few grammatical errors and undefined symbols:
* The authors should not use the capital letter after “:”, which is done throughout the paper.
* $\mathbb{I}$ and $\mathbb{S}$ are never defined in the text, and are probably typos, which should be simply replaced by I and S to indicate infected and susceptible individuals. Alternatively, the authors should maintain the notation by [1,2] and the one suggested by ICLR, which suggests using mathbb letters to identify sets (in this case the sets of susceptible, healthy, infected, quarantined, and deceased individuals).

## References
[1] Changjin Xu, Yicheng Pang, Zixin Liu, Jianwei Shen, Maoxin Liao, and Peiluan Li. Insights into covid-19 stochastic modelling with effects of various transmission rates: simulations with real statistical data from uk, australia, spain, and india. Physica Scripta, 99(2):025218, 2024.
[2] Abdullah S, Ahmad S, Owyed A H A, Aty E E, Mahmoude K and Shah H A 2020 Mathematical analysis of COVID-19 via new mathematical model, Chaos Solit Fractals 143 110585
[3] Wan, R., Zhang, X., \& Song, R. (2021, August). Multi-objective model-based reinforcement learning for infectious disease control. In Proceedings of the 27th ACM SIGKDD conference on knowledge discovery \& data mining (pp. 1634-1644).

**Questions:**

* How do your results differ qualitatively or quantitatively from the findings in Wan et al.? Do you arrive at different policy insights or confirm their findings at a larger scale?
 * What is the sensitivity of results to economic reward calibration? How was the reward function in line 219 selected?
 * Will code and data be made publicly available to enable comparison and reproducibility?
 * Why do the authors include just results for data from the UK in the analyses from pages 6 and 7 and do not include results for other countries in the Appendix?

---

### Official Review · Reviewer_zcvt · 2025-10-31

**Soundness:** 4
**Presentation:** 4
**Contribution:** 2
**Rating:** 4
**Confidence:** 3

**Summary:**

The paper designed a framework for evaluating intervention strategies during an infectious disease outbreak using multi-objective reinforcement learning (MORL) and stochastic differential equation (SDE). The SDE is used to simulate spread of infectious disease with added noise from the diffusion terms. Under the model, population is broken into 5 groups: susceptible, healthy/recovered, infectious, quarantined, and dead; the added diffusion reflects the stochasticity and variabilities in contact rate, infection waves, etc. that can happen during real outbreaks. A Pareto Conditioned Network RL agent is trained to optimize on three objectives: minimizing new cases, minimizing deaths caused by disease, and minimizing economic impacts. The action space is 3 dimensional, representing the 3 actions: quarantine, closure, and vaccination, with each action affecting the infectious disease dynamics (quarantined population can't infect susceptible population) and economic impact (closure and quarantine can have negative economic effects). Action scales from 0-10 representing the level of intervention. Simulation of SDE is simulate on a national-level using COVID-19 data in several countries. Compared against other existing models, namely agent-based simulation and the deterministic SIR model, the proposed SDE achieves the lowest AUC error rate (less than 1) while still capture the waves of infection during an outbreak. The trained RL agent is able to adapt to different strategies depending on which objective is preferred. If minimizing number of cases and deaths is preferred, then strategy results in lower cases while suffers economic disruption. Conversely, if economic impact is preferred, then strategy can result in uncontained outbreak while suffer minimal economic disruption. The proposed framework is also applied to other infectious disease with varying dynamics (polio, influenza, measles) and can be scaled to urban/city level simulation.

**Strengths:**

1. The SDE/MORL framework is scalable to different population level (national or urban) and is able to reflect stochasticity in infectious disease dynamics well compared to existing works in the field.
2. The framework is also adaptable to other infectious disease with varying dynamics (polio is spread through physical contacts and has higher mortality rate, influenza is airborne disease with low mortality rate, etc.). .
3. The SDE simulation is also compared against existing models in the field to show the effectiveness of the proposed framework, which is something that is not normally done in existing literature.
4. The RL agent is able to come up with strategy that minimizing new cases and minimizing economic impact, which realistically is good. The intervention level makes the result interpretable

**Weaknesses:**

1. The result is not exactly unique or unexpected. Of course it makes sense that prioritizing closure and quarantine will result in more economic disruption, and vice versa. The intensity level of the actions need to be defined better (for example, what does it means to take a closure action on level 1 vs level 10).
2. Intervention such as vaccination can be under a lot resource-constraint, realistically. Although the framework offers intensity level for the intervention action, it doesn't reflect the resource-constraint aspect of an ongoing outbreak.
3. Furthermore, these interventions are often targeted intervention (vaccine is prioritized for older population, quarantine for older individuals, etc.). The framework offers how intervention would look on a population level, but not the granularity and details required for an outbreak intervention.

**Questions:**

1. Regarding the level of intensity in intervention, why do they decide on the discrete level from 0-10, i.e. what does it mean to be level 5 closure? Does it mean closure on 50% of the population?

---

### Official Review · Reviewer_XfXj · 2025-10-31

**Soundness:** 3
**Presentation:** 3
**Contribution:** 1
**Rating:** 2
**Confidence:** 4

**Summary:**

Epidemics, such as the COVID-19 pandemic, have a disruptive socio-economic impact. Effective intervention strategies need to take into account these diverse aspects on top of disease containment. This makes it a natural use-case for multi-objective reinforcement learning (MORL), as the widespread availability of epidemiological statistics allow for reliable simulation frameworks. This work proposes a stochastic differential equation pandemic simulator, calibrated on infection data from 176 countries, on which they train a MORL algorithm (Pareto Conditioned Network (PCN)) to learn diverse intervention strategies with trade-offs on infections, deaths and economic impact. The authors then assess the quality of the learned strategies, showing that the MORL agent can navigate complex trade-offs between public health and economic stability.

**Strengths:**

- The problem tackled is a strong use-case with real-world impact
- The authors provide a thorough evaluation of the simulator, calibrated on infection data from 176 countries
- The algorithm learns different trade-offs, showcasing the benefits of MORL for assisting decision-making

**Weaknesses:**

To me, there is significant overlap between this work and [1]:
- both use a stochastic compartment model to simulate the epidemic.
- both use the PCN algorithm to learn optimal trade-offs of mitigation strategies. This work focuses on infections, deaths and economic impact, while [1] focuses on infections, hospitalizations and social impact.
- both qualitatively analyse the evolution of infections/deaths over time.
- both provide similar conclusions.

The main differences are:
- In this work, the agent can impact closure, vaccination and quarantine rates, at 10 different levels of granularity. Instead, [1] impacts closures of different social environments (work, school, leisure) with continuous levels of granularity.
- This works aggregates data over many countries, while [1] focuses on the first wave of COVID-19 in one specific country.
- [1] uses an age-structured compartment model with social contact matrices to model propagation rates while this work does not.
- This work extends the simulation to other diseases while [1] does not.

However, I do not believe these differences are significant enough to accept the paper in its current state.


[1] Reymond, M., Hayes, C. F., Willem, L., Rădulescu, R., Abrams, S., Roijers, D. M., ... & Libin, P. (2024). Exploring the Pareto front of multi-objective COVID-19 mitigation policies using reinforcement learning. Expert Systems with Applications, 249, 123686.

**Questions:**

I do not have specific questions.

---

### Official Review · Reviewer_1twF · 2025-11-01

**Soundness:** 2
**Presentation:** 3
**Contribution:** 2
**Rating:** 4
**Confidence:** 4

**Summary:**

This paper proposes a framework for learning pandemic intervention strategies using multi-objective reinforcement learning (MORL). The authors develop a stochastic differential equation (SDE)–based epidemic simulator that models five population compartments (susceptible, immune, infected, quarantined, deceased), and calibrate its transmission parameters using early COVID-19 trajectory data from 176 countries. Interventions (closure intensity, vaccination effort, quarantine rate) are treated as continuous control variables that directly modify the SDE transition dynamics. A Pareto-Conditioned Network (PCN) policy is then trained within this simulator to produce a set of Pareto-optimal policies that trade off epidemiological outcomes (infections, deaths) against economic disruption. The paper demonstrates how different priority weightings produce different intervention profiles, and extends the simulator to alternative pathogens (polio, influenza) and a measles vaccination-coverage case study to illustrate generality. The core contribution is a demonstration that MORL can generate interpretable intervention trade-offs when paired with a calibrated epidemic simulator.

**Strengths:**

1. Clear problem framing and motivation. The paper is explicitly motivated by a real and important challenge: balancing epidemiological and economic objectives when responding to pandemics. The motivation for using a multi-objective formulation is well articulated and conceptually appropriate.
2. Clarity and presentation quality are high. The paper is well written, logically structured, and visually clear. The figures effectively highlight the differences between intervention strategies under different objective preferences. The extension to polio, influenza, and measles helps convey the generality of the modeling workflow.
3. Simulator calibration is stronger than in many prior RL epidemic works. Unlike much of the reinforcement learning literature on pandemic control, the paper makes an effort to calibrate the underlying epidemic simulator using real historical data and to validate it against national case trajectories. This increases the plausibility of the modeled dynamics and improves interpretability relative to standard compartmental or purely synthetic models.
4. Use of Pareto-conditioned policies is a reasonable and interpretable approach to RL. Conditioning a single policy network on preference weights to approximate a set of intervention strategies is a meaningful and pragmatic application of multi-objective RL. Presenting trade-off curves and intervention profiles helps illustrate the structure of the policy space and makes the results accessible.

**Weaknesses:**

1. Lack of behavioral feedback and adaptive population response.
The simulator assumes homogeneous mixing, perfect compliance with interventions, and static contact behavior. In reality, pandemic dynamics are shaped by behavioral feedback: voluntary distancing in response to local prevalence, intervention fatigue, risk perception, and heterogeneous adherence. The authors acknowledge non-compliance as a "calibration issue" (lines 232-234), but this is insufficient—compliance varies dynamically in response to infection risk and policy stringency. Because the model does not capture endogenous behavior change or population heterogeneity, the learned policies reflect the dynamics of a compliant, homogeneous population rather than real adaptive systems. This fundamentally limits the external validity and transferability of the intervention strategies.

2. The use of reinforcement learning appears unnecessary and adds complexity without clear benefit.
The environment dynamics are fully known and differentiable, the state space is low-dimensional (5 compartments), the action space is small and discrete (3 interventions × 11 levels), and the objectives are explicit and smooth. There is little unknown environment to explore, the simulator is deterministic except for calibrated stochastic noise. Given these conditions, gradient-based trajectory optimization, model predictive control, or dynamic programming would be more natural and efficient. The choice of a Pareto-conditioned RL agent adds algorithmic overhead without demonstrating that RL is required, yields better solutions, or handles scenarios that simpler control methods could not.

3. No evaluation of learned policies beyond the simulator used for training.
The RL agent is trained and evaluated entirely within the same SDE simulator. There is no comparison to actual historical intervention outcomes, no off-policy evaluation using real data, no sensitivity testing to parameter misspecification and no assessment of whether learned policies align with documented strategies. As a result, it is unclear whether the learned policies would have any predictive or prescriptive value outside the model that generated them. The evaluation is internal consistency, not validation.

4. Economic cost model is overly simplified.
Economic impact is modeled as a linear function of closure intensity and quarantine proportion (lines 217-225), with an ad-hoc assumption that closure and quarantine are "equivalent when 20% of the population is infected" (line 224). This formulation is not grounded in empirical labor market data, sectoral analysis, mobility patterns, or microeconomic foundations. The resulting "economic trade-off" is therefore qualitative rather than quantitative, and the Pareto fronts (Figure 3) cannot be meaningfully compared to real-world economic costs such as GDP loss or unemployment. The authors acknowledge this limitation (lines 241-242: "economic cost is not calibrated with real-world costs"), but it fundamentally undermines the policy interpretation claims in the abstract and conclusion (lines 27-28: "support transparent, evidence-based policymaking").

5. Generalization demonstrations are qualitative rather than empirical.
The extensions to polio, influenza, and measles rely on parameter swapping rather than data-driven calibration or validation. The results show that intervention intensity changes when transmissibility changes — which is expected — but do not provide evidence that the learned policies correspond to real-world strategies or thresholds.

6. Limited engagement with modern scalable agent-based modeling approaches
The paper dismisses agent-based models as "computationally infeasible at country scale" (Appendix A.1, lines 628-630), but this assessment is outdated. Recent work has demonstrated that modern ABMs can efficiently simulate populations of 10⁶–10⁸ agents [1,3,7,8], use automatic differentiation for gradient-based calibration [2,7], incorporate learned agent behaviors via deep RL [4,5], neural networks [1,7], and LLMs [3], fit heterogeneous and streaming real-world data [1,2,7], and support rigorous counterfactual policy evaluation [3,6,9]. Further, These model have been explicitly validated to model disease and economic trade-offs for pandemic interventions [3, 6, 7].

These advances directly address the core limitations of this work: behavioral feedback, population heterogeneity, and out-of-distribution validation. The authors' dismissal contradicts their own claim that ABMs "capture fine-grained interactions" (line 72), and their justification that compartmental models "support integration of interventions" (line 75) applies equally to modern differentiable ABMs. At minimum, the paper should acknowledge this literature and provide a principled justification for choosing SDEs over ABMs, particularly given that ABMs naturally model the adaptive compliance and behavioral dynamics absent from this framework.

References for Weakness 6:
[1] A framework for learning in agent-based models (AAMAS 2024)
[2] Automatic Differentiation of Agent-Based Models (arXiv 2025)
[3] On the limits of agency in agent-based models (AAMAS 2025)
[4] The AI Economist (Science 2022)
[5] Learning and Calibrating Heterogeneous Bounded Rational Market Behaviour (AAMAS 2024)
[6] One-shot sensitivity analysis via automation differentiation (AAMAS 2023)
[7] Differentiable agent-based epidemiology (AAMAS 2023)
[8] Large Population Models (AgentTorch)
[9] Interventionally Consistent Surrogates for Agent-based Simulators (NeurIPS 2024)

**Questions:**

1. Comparison with modern agent-based models
Recent work shows ABMs scale to 10⁶–10⁸ agents with differentiable calibration and have been validated for pandemic intervention trade-offs [3,6,7]. Can you provide empirical comparison (runtime, sample efficiency, policy quality) with a modern differentiable ABM baseline, and justify why SDE is preferred despite lacking behavioral heterogeneity?

2. Necessity of reinforcement learning over direct optimization
Your simulator is differentiable with low-dimensional state/action spaces. Why is RL necessary rather than gradient-based trajectory optimization or model predictive control? Can you show that direct optimization fails to produce comparable Pareto fronts?

3. Economic cost calibration
Your cost function (lines 217-225) is ad-hoc and uncalibrated. Can you ground it in real economic data (GDP loss, unemployment) with interpretable units, or show policy rankings are robust to alternative cost formulations?

4. Out-of-distribution policy validation
All evaluation uses the training simulator. Can you provide: (a) comparison with actual historical interventions, (b) holdout evaluation on excluded countries. You could - show generality across waves of COVID-19 disease (e.g. delta vs alpha wave) or across countries by partitioning your 176 countries into 80/20 train/test splits, recalibrate σ on training countries only, verify simulator AUC error on test countries.

---

### Meta-Review · Area_Chair_kVLG · 2026-01-05

**Summary:**

The reviewers acknowledge the importance of the problem and the clarity of presentation, but identify several critical issues that motivate rejection. The proposed framework shows substantial overlap with prior work that already combines stochastic compartmental epidemic models with Pareto-conditioned multi-objective reinforcement learning, and the novelty beyond existing approaches is limited. The SDE-based simulator relies on some strong assumptions that significantly limits the realism and policy relevance of the learned strategies. Reviewers also question the necessity of reinforcement learning in this fully known, low-dimensional, differentiable setting, where simpler optimization or control methods would likely suffice.

**Reviewer Concerns:**

The authors did not submit a rebuttal.

**Reviewer Scores:**

In the absence of a rebuttal, reviewer scores would remain unchanged.

---

### Decision · Program_Chairs · 2026-01-26

Reject